



# A view on recent ice-nucleating particle intercomparison studies: Why the uncertainty of the activation conditions matters

Jann Schrod[1] and Heinz G. Bingemer[1]

[1]Institute for Atmospheric and Environmental Sciences, Goethe University Frankfurt, Frankfurt am Main, 60438, Germany

*Correspondence to*: Jann Schrod (schrod@iau.uni-frankfurt.de)

**Abstract.** Ice-nucleating particles (INPs) play a crucial role in cloud formation, influencing cloud phase, lifetime, and the onset of precipitation. Consequently, microphysical processes involving INPs strongly affect the radiative properties of clouds. However, when multiple INP counters measure simultaneously, notoriously high deviations between instruments in the range of 1 order of magnitude are commonly observed. These differences occur in ambient atmospheric measurements as well as in
laboratory studies. A regularly overlooked reason for these discrepancies may be related to uncertainties and errors in the temperature measurement. As the activation of INPs is a strong function of the nucleation conditions, relatively small inaccuracies in the temperature measurement may lead to significant over- or underestimations of the INP concentration. In this study, we have explored this effect as a potential reason for the differences observed among INP counters participating in 10 intercomparison studies that were published within the last 10 years. The stated temperature uncertainty of instruments
participating in these experiments ranged from ± 0.1 °C to ± 1.5 °C, and was most commonly specified as ± 0.5 °C. Potential deviations resulting from typical temperature errors were compared to the reported level of agreement among intercompared methods. As a measure of the potential INP error due to nucleation temperature error, we defined the error factor (*EF*) as the quotient of the ice nucleation activity at the actual nucleation temperature divided by the ice nucleation activity at a potentially erroneously measured temperature. Respective *EFs* were calculated for five distinct activation spectra based on four INP
parametrizations and one compilation of atmospheric INP data. *EF*s were between 1.1 and 3.2 for temperature errors of ± 0.5 °C, and between less than 2 and larger than 10 for temperature errors of ± 1.5 °C. *EF*s calculated from parametrizations of aerosols that are highly ice nucleation active were significantly larger than those derived from atmospheric data; although the effect was found to be still as large as a factor of 10 for certain temperature ranges in atmospheric activation spectra at a temperature error of ± 2 °C. When comparing two INP instruments, measurement biases may be of opposite direction, thus
resulting in expected differences of up to the product of both *EFs*. We found that opposite biases of +0.5 °C and −0.5 °C can therefore typically explain differences of a factor of 2, while opposite biases of +1 °C and −1 °C can theoretically explain differences of factors up to 5 or even 10, which is in the order of discrepancies typically reported in the literature on INP intercomparisons. These results highlight the need to carefully assess and report on uncertainties of the ice nucleation activation conditions.





## 1 Introduction

Ice-nucleating particles (INPs) have a significant effect on cloud microphysical processes, the formation of precipitation, cloud lifetime and the radiative properties of clouds (e.g., Mülmenstädt et al., 2015; Lohmann et al., 2016; Kanji et al., 2017). In the mixed-phase cloud (MPC) regime the existence of INPs shifts the relative phase composition from liquid water to ice, as supercooled droplets evaporate and activated ice crystals grow at their expense according to the Wegener-Bergeron-Findeisen

process (Wegener, 1911; Bergeron, 1928; Findeisen 1938). As a consequence, the lifetime of ice-containing MPCs is reduced compared to warm clouds, as ice particles precipitate out earlier due to the growth by subsequent condensation and riming (e.g., Pruppacher and Klett, 2010). Both the phase and lifetime of a cloud, in turn, influence the radiative properties of MPCs by decreasing the cooling effect when sufficient INPs are present (DeMott et al., 2010; Murray, 2017). In the cirrus regime, cloud icing is either initiated by heterogeneous or homogeneous ice nucleation depending on the availability of INPs and

ambient conditions of temperature ($T$) and ice supersaturation ($S_{ice}$). Heterogeneously formed cirrus clouds typically contain larger, but fewer ice crystals compared to homogeneously formed cirrus clouds. Because cirrus clouds have a net warming effect, less dense cirrus clouds, which were formed by INP activation, are considered to have a weaker warming effect (DeMott et al., 2010; Murray, 2017). However, the relative importance of INPs for cirrus formation is still largely unknown (Kanji et al., 2017, Krämer et al., 2021), as there are currently very few studies that attempted to measure the atmospheric concentration

of INPs below the homogeneous freezing limit (e.g., DeMott et al., 2003; Richardson et al., 2007; Wolf et al., 2020; Bogert, 2024).

INPs are operationally defined as particles that initiate heterogeneous ice nucleation, once characteristic conditions of supercooling and supersaturation are reached. Consequently, the measurement of INPs depend critically on these instrumental nucleation conditions, as well as on the detection and evolution of growing ice particles. These and other major measurement

challenges, combined with the rare nature and high variability of INPs, adds complexity. As a result, the typical uncertainty of INP measurements is unfortunately still substantial compared to many other fields within atmospheric sciences. It is not uncommon to find discrepancies of up to 1 order of magnitude or higher, when several different instruments, in parallel, observe the atmospheric INP concentration of ambient air or determine the ice nucleation activity of a known substance in the laboratory. This still holds true, even when the same aerosol material is prepared and generated in the same laboratory.

Recently, there have been a number of extensive endeavors to specifically intercompare INP instrumentation in the laboratory (e.g., DeMott et al., 2018) and in the field (e.g., DeMott et al., 2024, Lacher et al., 2024), resulting in mixed levels of agreement (see Sect. 2).

When evaluating the degree of consistency in instrument intercomparisons, the community mostly ascribes a reasonable or good agreement, if INP concentration measurements fall within a factor of 10, or if the temperature spectra of two or more

measurements generally follow a similar trend. However, what is often missing in these assessments is an evaluation of the implications of observed differences between instruments for modeling. Specifically regarding how accurate and consistent





measurements need to be in order to meaningfully improve the representation of cloud microphysical processes, precipitation and radiative interactions in models (DeMott et al., 2024).

There are various reasons that might explain the differences in the INP concentration observed during an intercomparison,
which may highly depend on the specific circumstances of the measurements. However, the disagreement between instruments usually arises from differences in aerosol sampling, from a mismatch in sampling time, and most seriously from incomplete overlap or inaccuracies in the instrumental activation conditions. For example, DeMott et al. (2018) points to temperature uncertainty being a key factor influencing the observed differences in INP concentration during the large-scale laboratory intercomparison of FIN-02 (Fifth International Workshop on Ice Nucleation – phase 2). However, the effect of temperature
uncertainty may not have been fully considered in several other studies thus far and deserves a more thorough investigation.

The number of activated INPs is known to be very sensitive to the activation conditions, i.e. the temperature for immersion freezing, and additionally the ice supersaturation, when measurements are performed below water saturation. This fact is of course well-established and is implemented in parametrizations of specific aerosol species (e.g., Atkinson et al., 2013; Hiranuma et al., 2015; Wex et al.; 2015; Ullrich et al., 2017; Hiranuma et al., 2019) and natural atmospheric aerosol (e.g.,
Fletcher, 1962; DeMott et al., 2010, DeMott et al., 2015). Compilations of INP observations as presented for example in Kanji et al. (2017) or Petters and Wright (2015) also feature this finding prominently, showing a distinct exponential increase in the INP concentration with decreasing nucleation temperature. Per 5 °C cooling an increase in the INP concentration by factors between 2 and up to ~50 can be observed in the atmosphere. Specific ice-active aerosols may even cause an increase of the INP activity on the order of a factor of up to ten to a hundred thousand for biological material like Snomax or by a factor of a
couple of hundreds for the highest active mineral aerosol (i.e., K-Feldspar) per 5 °C cooling. Consequently, comparably small changes in the activation conditions can lead to significant changes in the number of activated INPs. Therefore, if a bias $\delta T$ or $\delta S_{ice}$ exist between the true nucleation conditions $T_n$ and $S_{ice,n}$ and the erroneously measured instrumental conditions $T_m$ and $S_{ice,m}$, respectively, the INP concentration may be substantially over- or underestimated. The relation between the actual nucleation conditions, the assumed nucleation conditions, which are either directly measured or calculated in some way, and
the error in the nucleation conditions can be formulated according to Eq. (1) and Eq. (2):

$$T_n = T_m + \delta T \,, \tag{1}$$

$$S_{ice,n} = S_{ice,m} + \delta S_{ice} \,. \tag{2}$$

The uncertainty in the activation conditions of INP instruments are sometimes only stated briefly in the method section of respective publications, or in the supplementary material, if at all. Furthermore, a majority of figures showing results from INP
measurements only entail error bars of the ordinate, representing the uncertainty of INP concentration ($n_{INP}$), frozen fraction ($FF$), activated fraction ($AF$), active site density by surface ($n_s$) or mass ($n_m$). The uncertainty in activation temperature and ice supersaturation is often not considered in the estimate of the INP uncertainty. However, errors in the activation conditions likely outweigh other factors contributing to the INP concentration uncertainty in many cases, as we will discuss in the following sections. The alternative would be to clearly indicate the activation uncertainties as separate temperature error bars



of the abscissa. However, these are often missing in figures either in favor of clarity or because they are inadvertently misjudged as unimportant.

In this study, we will focus on the magnitude, nature and a potential cause of observed discrepancies between INP instruments running in parallel by investigating the role of inherent instrument uncertainties in the activation conditions. First, we will review the literature of INP intercomparison measurements and compile the temperature uncertainties stated therein. Then we

will estimate the effect of this temperature inaccuracy on the INP activity in relation to the differences actually observed in the intercomparisons.

## 2 Reviewing recent INP instrument intercomparison measurements

We considered a study for our investigation if the following criteria were all met:

(1) The study was published within the last 10 years.

(2) The study presents either measurements of atmospheric INPs or investigates the ice nucleation activity of specific aerosol species in a laboratory setting.

(3) At least four individual instruments were used to determine the INP properties simultaneously, regardless of whether an intercomparison of instruments was set out as a prime objective of the study or not.

We found 10 studies that met these criteria (Table 1). Since the literature research was conducted through a comprehensive

review of available sources rather than by systematic use of specific search terms, it is possible, although rather unlikely, that we inadvertently missed a few studies that would have otherwise met the criteria. Table S1 lists the identified studies, the participating INP instruments and their stated uncertainties in nucleation temperature ($\Delta T$). If uncertainties were not reported in the original study or their associated supplementary material, distinctly referenced instrument papers were browsed for this information, assuming the uncertainty assessment did not change. If the uncertainty of a method was also not reported in the

methods referenced technical description, other sources such as intercomparison studies, in which the method participated, were consulted. If no uncertainty was reported in the original study and there were deviating temperature uncertainty assessments from different referenced studies, the latest data was taken into account. When in doubt, the highest uncertainty reported was listed. Assessments of principally identical commercially available instruments (e.g., SPIN) of different institutes were assumed to be valid for other versions. Note, that the uncertainties stated in the studies at the time of publication may or

may not reflect the current uncertainty of a specific instrument. If a specific instrument partook in multiple intercomparison studies, all entries are listed.

From the ten identified studies a total of 104 temperature uncertainties are documented from 43 individual instruments. In more than one fifth of the cases (22/104) no temperature uncertainty was indicated in the original study or its supplementary material. A number of uncertainty estimates were not stated in absolutely unambiguous terms, but needed some interpretation.



Furthermore, six studies did not have any figures showing the temperature error, while the other four studies had only a minor subset of figures showing the temperature error.

While we naturally have confidence that groups carefully assess their respective temperature uncertainty to the best of their ability, most times uncertainty statements lacked a thorough description. More crucially, some estimates seemed to only account for sensor accuracy, while other factors relating to the total uncertainty were possibly not or not fully considered.

Sometimes additional sources of temperature uncertainties were mentioned in writing, but no corresponding value was attributed.

**Table 1: Identified INP intercomparison studies from 2015 to 2024.**

| Study | Laboratory / Ambient | Aerosol type / Location | Number of participating instruments |
|---|---|---|---|
| Hiranuma et al., 2015 | Laboratory | Illite NX | 17 |
| Wex et al., 2015 | Laboratory | Snomax | 7 |
| Burkert-Kohn et al., 2017 | Laboratory | Microcline, Kaolinite, Birch pollen | 4 |
| DeMott et al., 2017 | Ambient | Western USA | 5 |
| DeMott et al. 2018 | Laboratory | Illite NX, K-Feldspar, Argentinian Soil Dust, Tunisian Soil Dust, Snomax | 21 |
| Hiranuma et al., 2019 | Laboratory | Cellulose MCC, FC, NCC | 20 |
| Knopf et al., 2021 | Ambient | Southern Great Plains, USA | 6 |
| Brasseur et al., 2022 | Ambient | Southern Finland | 6 |
| Lacher et al., 2024 | Ambient | Central France | 10 |
| DeMott et al., 2024 | Ambient | Rocky Mountains, USA | 6 |

For online processing chambers (e.g., continuous freezing diffusion chambers (CFDCs), the portable ice nucleation experiment

(PINE)) experimental uncertainties in the activation conditions may generally be related to accuracy, drift, precision and placement of the temperature (and humidity) sensor, and to the spatial and temporal variation of temperature (and humidity) within the chamber and during an experiment. For CFDCs the spatial variation in the wall temperatures in relation to the resulting temperature profile in the laminar flow region seems to be the main factor of uncertainty. For example, in the new PINCii instrument 58 type-K thermocouples measuring at 1 Hz were specifically implemented to better resolve the spatial

inhomogeneity of the temperature (Castarède and Brasseur et al., 2023). Despite their rigorous monitoring efforts combined with a high number of coolant injection points in order to minimize temperature inhomogeneity, they concluded that improvements are still necessary. Garimella et al. (2017) highlighted that for CFDCs the fraction of particles in the laminar flow is often lower than theorized, and thus particles may be exposed to varying temperatures and humidities, which may be in strong contrast to the predicted activation conditions. DeMott et al. (2015, 2017 and 2018) also raised this issue and surmised



that the introduction of the air flow into the CFDC due to the specific instrument design could result in particles outside the laminar flow region, where temperature and humidity vary. Garimella et al. (2017) estimated that this may lead to systematic underestimation of the INP concentration by factors between 1.5 and 9.5, because in reality not all particles are activated to droplets, in contrast to what the assumed activation conditions would suggest.

     For the cloud expansion-type chamber PINE the inhomogeneous temperature distribution during an expansion is identified as

the main contributor to the temperature uncertainty (Möhler et al., 2021). In an example experiment presented by Möhler et al. (2021) (their Fig. 6) three gas temperature sensors record a difference of about 2 °C from the warmer top to the colder bottom of the chamber during the start of an expansion. As the experiment progresses the deviation in temperature in the chamber further increases to up to 4 °C at the end of the expansion. Usually the coldest sensor at the bottom is assumed to be representative for the activation conditions and is therefore used when results are presented (e.g., Möhler et al., 2021, Knopf

et al., 2021), which is a reasonable, but ultimately unverified assumption. When the cirrus regime is investigated in PINE-like instruments (i.e., PINEair, under construction for aircraft use) a step-wise rapid expansion is initiated by use of a buffer volume, followed by a continuous expansion (Bogert, 2024). It is observed that under these experimental settings the measured temperature significantly deviates from the calculated near-adiabatic temperature profile, as the sensors do not react fast enough to the adiabatic expansion and thus cannot be used to characterize the activation conditions (Bogert, 2024). Because

the question of heterogeneous vs. homogenous freezing is of central importance in the cirrus regime, knowing the actual nucleation temperature and as a result the ice supersaturation is key to distinguish the freezing mechanism. In this regard, temperature uncertainties will significantly affect the uncertainty of the ice supersaturation in this low temperature regime.

     Please note that the uncertainty of $S_{ice}$ was not examined in detail in this study, because most results from the identified studies investigated immersion freezing, where the relative humidity should be firmly above water saturation (with the exception of

CFDCs not reaching an equivalent humidity needed to immerse all particles in some cases). However, when ice nucleation below the water saturated regime is investigated by deposition nucleation in the MPC or cirrus regime, the uncertainty of $S_{ice}$ may play a significant role in the overall instrument uncertainty. When reported, the range of stated uncertainties of $S_{ice}$ was between 1% and 5% relative humidity in the identified studies.

     Droplet freezing cold stage (DFCS) instruments have ultimately the same general causes of temperature uncertainties as online

processing chambers. The error in nucleation temperature depends on how accurate and how representative the measured temperature is. As surface temperature measurements (e.g. by infrared radiation) are typically more uncertain than spot measurements, mostly one or multiple point measurements are performed using various kinds of temperature sensors on the surface of or within the cold stage apparatus. Therefore, it is not straight-forward to estimate the freezing temperature of an individual droplet, as the temperature measured may or may not be representative for the exact location on the cold stage where

the freezing event occurred. The temperature homogeneity of a cold stage may possibly also depend on the cooling rate, which may vary between some tenth °C/min and up to 10 °C/min. Heat exchange and latent heat released by freezing droplets may also affect the temperature measurement and the freezing behavior of other droplets. Additionally, the time resolution of the algorithm that identifies the freezing events from images or video, should be considered for the temperature uncertainty. In the




identified studies this time resolution was mentioned in some cases, but was not included in the uncertainty estimates in Table

S1, as it was not explicitly stated by any of the researchers. For example, DeMott et al. (2017) stated that NIPR-CRAFT analyzed video images at 0.5 °C intervals to determine the number of frozen droplets, while the indicated temperature uncertainty was only ± 0.2 °C.

Figure 1a and 1b present the temperature uncertainty for individual instruments for each unique temperature assessment as a bar and box plot, respectively. When the uncertainty statement for a specific instrument changed over time, multiple bars or

circle symbols are depicted in Fig. 1a and 1b, respectively. Similarly, multiple bars (or circle symbols) are shown for each physical copy of the same instrument, if they are affiliated with different institutes. The reported uncertainties for offline processing instruments (purple colors) tends to be lower (usually between ± 0.2°C and ± 0.5 °C) than those from the more complex online processing instruments (blue colors), for which stated temperature uncertainties were typically between ± 0.5°C and ± 1.5 °C. The median temperature uncertainty reported was ± 0.5 °C.

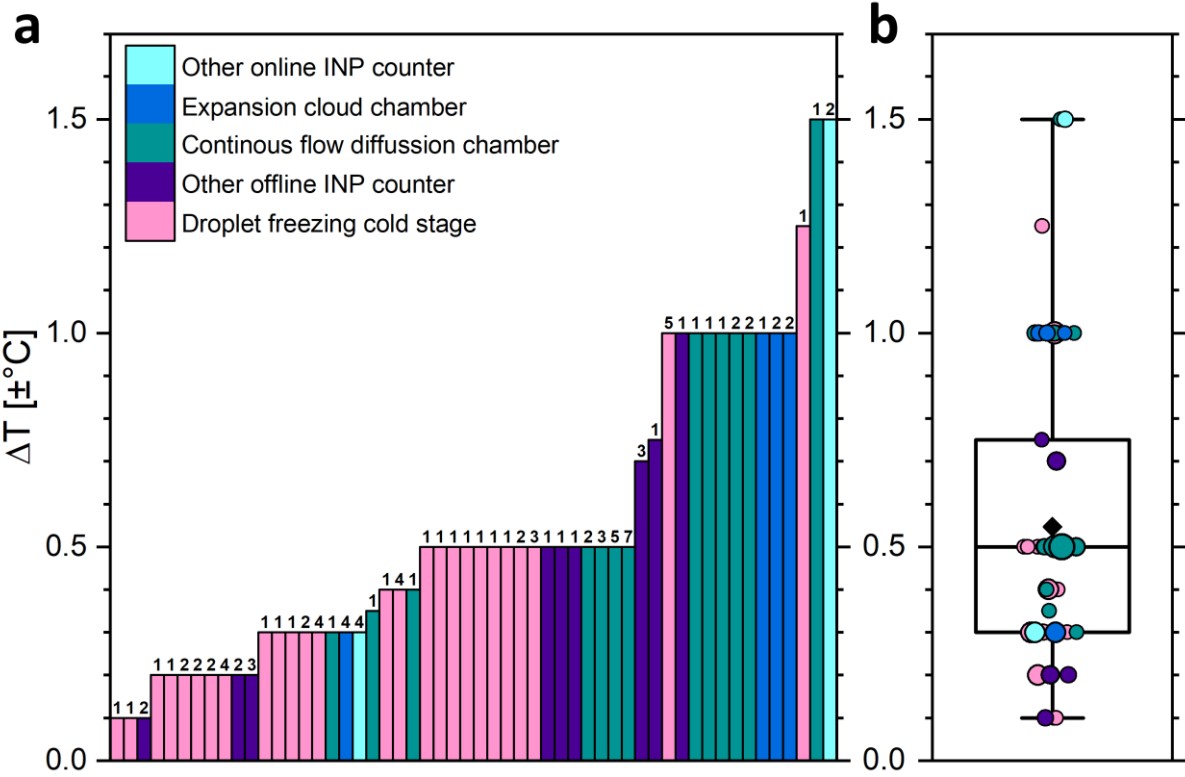


**Figure 1: Individual temperature uncertainty statements of instruments participating in the intercomparison studies as identified in Table 1, represented as bars (a) and circle symbols (b). Numbers and circle symbol sizes indicate how often an uncertainty assessment of an individual instrument was reported. The box plot indicates the range, 25% percentile, median, 75% percentile and average (black diamond) of the stated temperature uncertainties (b).**

Table 2 summarizes the general quantitative assessment of the level of agreement between INP measurements of different instruments in the studies considered. We will outline the main findings in the following paragraph, while the reader is referred





to Table 2 for the detailed statements. Most of the intercomparison results indicate that INP measurements of different instruments are usually agreeing within 1 order of magnitude. Comparisons of atmospheric INP measurements are generally not better or worse than laboratory intercomparisons. DeMott et al. (2024) also comes to this conclusion, when comparing the

results of the FIN-03 field study to the FIN-02 laboratory intercomparison (DeMott et al., 2018). They conclude that ambient INP concentrations can be measured with a similar accuracy as laboratory experiments, despite the variability in aerosol composition, concentration and size distribution. The atmospheric intercomparison that likely yielded the closest agreement of INP concentrations was described by Lacher et al. (2024), where 10 INP instruments showed differences usually within a factor of 5. For some 10 percent of the time the agreement was even as good as within a factor of 2, depending on the

combination of instruments considered. Judging from the literature available, this seems to be the best that is achievable with the current instrumentation and measurement uncertainties. The agreement of laboratory intercomparison varied markedly with the specific material investigated, ranging from differences that were in parts within the measurement uncertainties (Wex et al., 2015; Burkert-Kohn et al., 2017) to discrepancies exceeding more than 3 or 4 orders of magnitude (Hiranuma et al. 2015, 2019). Additionally, the diversity in aerosol preparation, generation and sampling, and the specific measurement procedure

had a large influence in some studies (Hiranuma et al., 2015, 2019). Encouragingly, overall there seems to be a trend towards a higher level of agreement in the more recent intercomparison studies compared to those from a decade ago.

Table A1 expands upon the pure assessment of the intercomparison of Table 2 by listing various study-specific findings regarding temperature regions, where the differences were most pronounced (i.e., usually the warm and cold end of the data), identified instrumental or sampling-related reasons for discrepancies, or the unique ice-nucleating characteristics of the

investigated aerosol, which may influence the overall level of agreement. As we will explore in more detail in Sect. 3, the specific slope of the temperature dependency was identified as one main cause for observed differences in some studies.

**Table 2: Level of agreement among INP methods during the intercomparison studies.**

| Study | Level of Agreement |
|---|---|
| Hiranuma et al., 2015 | • Differences can reach up to 3 orders of magnitude at the same temperature or 8 °C at the same $n_s$ |
| Wex et al., 2015 | • 72% of data points were within 1 °C at the same $n_m$; 78% were within 2 °C<br>• At the high end of $n_m$ all data were within a factor of 3 |
| Burkert-Kohn et al., 2017 | • Differences can exceed a factor of 3 in the water supersaturated regime<br>• Immersion mode measurements agreed within the experimental uncertainty for all aerosol types; differences in deposition and condensation mode measurements were up to 2 orders of magnitude |
| DeMott et al., 2017 | • Differences were mostly less than 1 order of magnitude, while at ~1 INP/L differences can reach nearly 2 orders of magnitude<br>• CFDCs were often 2 to 5 times lower than DFCSs |




| Study | Level of Agreement |
|-------|--------------------|
| DeMott et al., 2018 | • Typically, differences were within 1 order of magnitude, but varied with aerosol type<br>• Snomax showed the best agreement overall; at <−10 °C differences were as low as a factor of 5<br>• Differences were within 1 order of magnitude for Argentinian soil dust; Tunisian soil dust showed similar, but slightly worse agreement<br>• K-Feldspar and Illite NX showed discrepancies partially exceeding 2 – 3 orders of magnitude |
| Hiranuma et al., 2019 | • Differences can reach up to 4 orders of magnitude at the same temperature or 10 °C at the same $n_s$ |
| Knopf et al., 2021 | • INP concentrations seem to be usually within 1 order of magnitude[a] |
| Brasseur et al., 2022 | • INP concentrations were usually within 1 order of magnitude and followed similar trends, with one CFDC being consistently lower by a factor of 10 |
| Lacher et al., 2024 | • Online INP counters had 35% and 70% of data within a factor of 2 of a reference CFDC; 80% and 100% were within a factor of 5, respectively<br>• DFCSs from the same inlet had on average 45% of data within a factor of 2 of a reference DFCS; 77% were within a factor of 5 |
| DeMott et al., 2024 | • Average INP concentrations agreed within factors ranging from nearly 1 to 5.5, corresponding to differences of 3.5 °C to 5 °C; Depending on the instruments compared, 60 % to 100% of individual data were within 1 order of magnitude<br>• Differences can increase to up to 2 orders of magnitude between −20 °C and −25 °C |

**a**: The intercomparison of different INP instruments was not discussed in the Knopf et al. (2021), because the focus was to check for closure between aerosol measurements and ice formation. This assessment is based on the example data of the afternoon of October 15, 2019 presented in their Fig. 4b and Fig. ES2.

## 3 Estimating the effect of the temperature uncertainties

We refrain here from quantitatively comparing the effect of the uncertainty of the nucleation conditions to the stated uncertainties in the INP concentration for individual instruments, because the latter are usually not expressed simply as percentage or concentration range, but depend on measurement specifics. Thus, this would go beyond the scope of this study. However, the differences observed among INP measurements often exceed the reported uncertainties or error bars of the instruments considerably. As shown, these discrepancies can be on the order of 1 magnitude or more, while the INP concentration uncertainty of individual instruments, often related to freezing statistics or the counting process of ice particles, is usually only on the order of several tens of percent.





To estimate how errors in nucleation temperature propagate to uncertainty of INP measurements, we will in the following define an error factor (*EF,* see Eq. 7 below), and derive *EFs* for five empirical data functions that relate INP activity to
nucleation temperature. These functions include a compilation of atmospheric measurements by Petters and Wright (2015, referred to hereafter as PW15), as well as four INP parameterizations (see Fig. B1). The parametrizations were either derived from continental measurements of the atmospheric INP concentration (DeMott et al., 2010, hereafter D10), or from laboratory and field measurements of mineral dust (DeMott et al., 2015, hereafter D15), or from the ice nucleating activity of known ice active biological material (Snomax, Wex et al., 2015, hereafter W15) and mineral components (K-Feldspar, Atkinson et al.,
2013, hereafter A13). For D10

$$n_{\mathrm{INP,D10}} = a \, (273.16 - T_K)^b \, (n_{a>0.5})^{(c(273.16 - T_K) + d)} \, , \tag{3}$$

where a = 0.0000594, b = 3.33, c = 0.0264, d = 0.0033, $T_K$ is the temperature in Kelvin, and $n_{a>0.5}$ is the aerosol number concentration of particles with a diameter larger than 0.5 μm [scm$^{-3}$] and $n_{\mathrm{INP}}$ is the resulting INP concentration per standard liter. For D15

$$n_{\mathrm{INP,D15}} = (cf) \, (n_{a>0.5})^{(\alpha(273.16 - T_K) + \beta)} \, e^{(\gamma(273.16 - T_K) + \delta)} \, , \tag{4}$$

where α = 0, β = 1.25, γ = 0.46, δ = −11.6 and the calibration factor *cf*, which is suggested to be set to 3 for atmospheric applications in order to emulate the influence of measuring the maximum immersion freezing concentration versus CFDC measurements at $S_{\mathrm{water}} = 105\%$ (*cf* = 1). For W15

$$n_{m,W15} = 1.4 \times 10^9 \times \left(1 - e^{(-2 \times 10^{-10}) \times e^{(-2.34 \times T_C)}}\right), \tag{5}$$

where $T_C$ is the temperature in °C and $n_m$ is the active site density by mass [mg$^{-1}$]. For A13

$$n_{s,A13} = e^{(-1.038 \times T_K + 275.26)} \, , \tag{6}$$

where $n_s$ is the active site density by surface [cm$^{-2}$].

In case of PW15, we used the average of the base 10 logarithm of the upper and lower end of the data envelope in order to investigate a temperature spectrum that is representative for the compilation of precipitation samples (red line in Fig. B1c,
compare Petters and Wright, 2015).

In the following analysis we present the error factor *EF* of $n_{\mathrm{INP}}$, $n_m$ or $n_s$ at the actual nucleation temperature $T_n$ relative to the potentially falsely assumed temperature $T_m$ due to the inaccuracy of the temperature measurement, as a function of $T_m$ (Fig. 2 and Table 3):

$$EF = \frac{n_{i,j} \, (T_n)}{n_{i,j} \, (T_m)} \, , \tag{7}$$



where the index *i* stands either for *INP*, *s* or *m*, defining the INP concentration, active site density by surface or mass to be compared according to Eq. (3) to (6) or the red line in Fig. B1c, and the index *j* stands either for D10, D15, W15, A13 or PW15.

When $\delta T$ is greater than 0, *EF* is less than 1, as indicated by the blue colors in Fig. 2. This means the ice nucleating ability is underestimated, because the actual nucleation temperature was warmer than the inaccurately measured temperature. To
illustrate this, let's consider a hypothetical scenario following Fig. 2d, where *EF* is equal to 0.5 (upper grey line) at a $\delta T$ of approximately +1.5 °C. Given this $\delta T$, a hypothetical instrument mistakenly assuming to measure at −25 °C would actually report $n_{\mathrm{INP}}$ of the true temperature of −23.5 °C. At the incorrect temperature reading, the instrument registers a $n_{\mathrm{INP}}$ of 25 L$^{-1}$. However, if the temperature had been accurately measured at −25 °C, the concentration should have been 50 L$^{-1}$ (at $n_{a>0.5}$ = 10 cm$^{-3}$). When $\delta T$ is less than 0, *EF* is greater than 1 (red colors in Fig. 2), respectively, and the ice nucleation ability
is overestimated. For simplicity, in the following paragraphs we discuss only *EFs* > 1, while the (near-) symmetrical results are similar or the same for the inverse of that factor (*EFs* < 1, i.e., when $\delta T$ has the same absolute value, but a different sign).

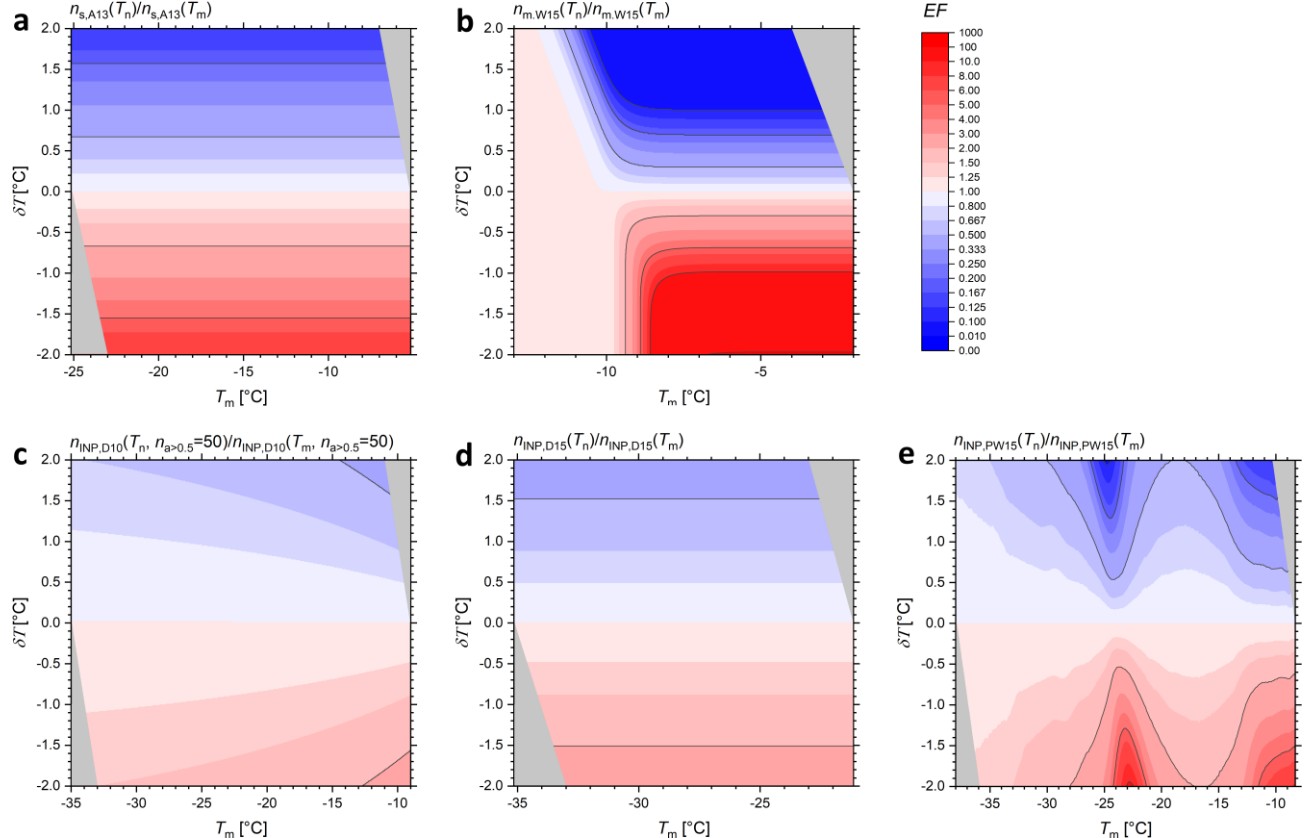

**Figure 2: Error factors of A13 (a), W15 (b), D10 (c), D15 (d) and PW15 (e) as function of assumed measurement temperature $T_{\mathrm{m}}$ and temperature error $\delta T$. When visible, grey isolines show the constant *EFs* of 2, 5, 10 and 100 (and their inverse). For D10 $n_{a>0.5}$**
**is set to 50 cm$^{-3}$. Grey triangle overlays confine the valid temperature range of the original data function.**



For the laboratory-based parametrizations A13 and W15 of the highly ice-active materials K-Feldspar and Snomax, respectively, a very strong temperature dependency can be observed. In the case of A13, this translates to *EFs* exceeding a value of 2 at a temperature error of ± 0.7 °C, and 5 at a temperature error of ± 1.5 °C, independent from the assumed measurement temperature $T_m$ (Fig. 2a). For Snomax $n_m$ reaches a maximum plateau at approximately −10°C (Fig. B1b),

therefore the *EF* is equal to 1, when $T_m - \delta T$ is equal to this temperature or lower. However, in the temperature range from −2 °C to −10 °C the relative *EF* increases strongly within the $\delta T$ window. A factor 2 is seen at ± 0.3 °C, at ± 0.7 °C the *EF* exceeds 5, at ± 1 °C the *EF* results in an over- or underestimation of 1 order of magnitude, and at ± 2°C the error can even reach up to a factor of 100 (Fig. 2b).

Atmospheric INP temperature spectra are in general never as steep as those derived from A13 or W15. Accordingly, we find

that for D10 the *EF* is for most of the temperature spectrum below a factor of 2 even at a $\delta T$ of ± 2°C (Fig. 2c). Fig. B2 shows the sensitivity of the *EF* of D10 to varying $n_{a>0.5}$, which is generally low. D15, which is representative of a more dust-rich environment, still only exceeds an *EF* of 2 when $\delta T$ larger than ± 1.5°C (Fig. 2d). However, atmospheric INP temperature spectra may not always increase strictly uniform with decreasing temperature, as can be seen for PW15 (red line in Fig. B1c), because the atmospheric aerosol is a mixture of different species that are ice-active at different temperatures. In our analysis,

the average of the precipitation samples of PW15 exhibits a strong increase in INP activity with temperature at the warm end of the spectrum of around −10 °C (likely biological INPs), as well as between −20 °C and −25 °C (likely mineral INPs). Therefore, the measurement of atmospheric samples may be more susceptible to errors in the temperature measurement in these temperature ranges. For example, at $T_m = -23$ °C an *EF* of 2 is observed at ± 0.6 °C, the factor increases to 5 at ± 1.3 °C, and is larger than 10 at ± 2 °C (Fig. 2e). DeMott et al. (2024) pointed out that the INP variations between instruments measuring

at Storm Peak, CO, USA significantly increased for the −20 °C to −25 °C range to nearly 2 orders of magnitude, further demonstrating the point. A similar result was also observed by Lacher et al. (2024) for the intercomparison at Puy de Dôme, France.

If temperature errors are kept at or below ± 0.5 °C, the resulting error in the INP activity is acceptable, reaching only up to an *EF* of 1.7, when W15 is excluded (Table 3). If temperature errors however exceed ± 1°C, the reported INP activity can be

considerably biased. At ± 1 °C errors translate to inaccurate INP estimations by a factor of 1.3 to 3.3 for atmospheric spectra. At ± 2 °C the error may result in over- or underestimation of up to or more than 1 order of magnitude depending on the activation spectrum of the analyzed sample.



**Table 3: Maximum relative error factor resulting from four example temperature errors $\delta T$ at example measurement temperatures $T_\mathrm{m}$. Note that the *EF* in A13 and D15 is independent of $T_\mathrm{m}$. Similarly, the *EF* in D15 is also independent of $n_{\mathrm{a}>0.5}$.**

|  | $\delta T = 0.2\ °C$ | $\delta T = 0.5\ °C$ | $\delta T = 1\ °C$ | $\delta T = 2\ °C$ |
|---|---|---|---|---|
| A13 | 1.2 | 1.7 | 2.8 | 8.0 |
| D10 ($n_{\mathrm{a}>0.5} = 50\ \mathrm{cm}^{-3}$, $T_\mathrm{m} = -25\ °C$) | <1.1 | 1.1 | 1.3 | 1.6 |
| D15 | 1.1 | 1.3 | 1.6 | 2.5 |
| PW15 ($T_\mathrm{m} = -25\ °C$) | 1.2 | 1.6 | 3.3 | 10.2 |
| W15 ($T_\mathrm{m} = -7\ °C$) | 1.6 | 3.2 | 10.4 | 107.6 |

It is important to note that the results presented so far considered only the divergence between one instrument's measurement and the "true" value according to the assumed data function. However, when two or more INP instruments are compared, each instrument may diverge from the "true" INP concentration due to individual temperature errors. For example, if one instrument assigns a +0.5 °C high bias in nucleation temperature and another instrument assigns a −0.5 °C low bias, the total combined error of one instrument against the other is the product of both *EFs*. Therefore, a factor of 2 difference may appear between two instruments, when the *EFs* are each equal to 1.41, which is well within the range of calculated *EFs* for ± 0.5 °C. A factor of 5 difference between two instruments could be expected at *EFs* of 2.24, which is well within the range of calculated errors for ± 1°C. Even a factor of 10 difference between two instruments (i.e., the square of an *EF* of 3.16) could be explained by erroneous temperature measurements of about ± 1 °C, depending on the aerosol activation spectrum.

Therefore, in order to keep divergences among different instruments low, efforts should be made to limit errors in the measurement, calculation or assessment of the nucleation temperature to significantly below 1 °C, and best below 0.5 °C.

## 4 Conclusions

The role of uncertainties and inaccuracies of the activation condition of experimental ice nucleation studies is often underexplored, when results are interpreted or INP concentrations from multiple instruments are compared. This is evident in the lack of temperature error bars in many figures of published INP literature. Similarly, in 21% of all cases no temperature uncertainty was reported in the primary literature of the INP intercomparison studies that were analyzed here. Overall, temperature uncertainty assessments of researchers varied from ± 0.1°C to ± 1.5 °C, with a median of ± 0.5 °C.

In principle, the assumed nucleation temperature can diverge from the actual nucleation temperature due to inaccurate or imprecise measurements, measurements that are not representative for the region of ice formation due to spatial or temporal inhomogeneities in the instrument, insufficient spatial coverage or non-optimal placement of sensors, sensor drift, or uncertainties and other unknown effects contributing to errors when calculating the nucleation conditions in some way. Beyond the uncertainties of the temperature measurement, there are several other, potentially large, uncertainties for offline INP



sampling that are related to performing the background freezing of pure water, and the handling and storage of samples in
general, which naturally need to be considered as well (e.g., Polen et al., 2018; Beall et al., 2020).

In studies that intercompared at least four INP counters, we found differences between instruments to be most often in the range of 1 order of magnitude. On the extreme, in some studies differences in the range of 3 or 4 orders of magnitude were experienced, likely due to specific aerosol properties and the way the aerosol was generated and processed (Hiranuma et al., 2015, 2019). Sometimes variations were as low as a factor of 2 to 5, while only rarely and usually only under controlled
laboratory conditions, agreement between subsets of instruments was within the actual experimental uncertainty (in terms of INP concentration error). Considering these findings, the disconnect between stated INP concentrations uncertainties, which are often in the range of some 10 percent, and the observed differences in instrument intercomparisons is glaring.

As the results presented here indicate, potential errors in temperature measurements may significantly contribute to the observed deviations. It is difficult to assess, how much of the differences in the identified intercomparisons are exactly related
to this effect. It very much depends on the steepness of the activation temperature spectrum of the specific material investigated or the aerosol composition of the atmospheric sample, and the actual error in the temperature measurement of participating instruments.

For typically reported temperature uncertainties the here analyzed temperature error effect can be as small as a factor of 2 or less for continental atmospheric samples that do not show the strong signature of mineral dust or biological particles (i.e., those
that behave like D10). Still, there may be temperature ranges in atmospheric samples that show a distinct temperature dependency, thus *EFs* of 5 or greater can be possible due to misjudging the nucleation temperature. The expected errors are by far the greatest in laboratory intercomparisons with highly ice active materials.

The strong temperature dependence of specific ice-nucleating materials combined with measurement errors of nucleation temperature were in fact considered as a partial reason for the large discrepancies observed among individual instruments in
some intercomparison studies. For example, it is explicitly stated that the propagating temperature uncertainty dominates the variation of the resulting INP concentration uncertainty for the CSU-CFDC, which was estimated to be ± 60 % for any temperature (Hiranuma et al., 2015). While ± 60 % is a reasonable assumption for not particularly active atmospheric samples, we have demonstrated here that errors can be significantly larger. Hiranuma et al. (2015) concluded that biased overall accuracy and precision of instruments can be related to factors shifting the activation temperature, when discussing potential reasons for
observed diversity in the intercomparison of Illite NX. Furthermore, DeMott et al. (2018) consistently noted that discrepancies in the INP concentration between different instruments increased in regions where the investigated aerosol material showed a stronger temperature dependency, and attributed uncertainties in temperature measurements as a key reason for this finding. DeMott et al. (2017) suggested that a temperature offset by at least 1 °C for one DFCS related to errors in the droplet temperature measurement may explain why this method generally showed overall higher INP concentrations. Following this
example, researchers should always pay close attention to systematic differences between instruments, which can indicate methodical biases in the activation condition measurements. If unequivocally identified, such systematic biases may potentially be corrected post-measurement, and should be separated from unsystematic variations between instruments. However, even



when a general systematic bias between instruments is identified, there still can be measurement periods where the systematic bias disappears or even reverts (e.g., DeMott et al., 2024).

While these examples show that the community is generally aware of the temperature error effect, no study actually tried to quantitatively estimate the contribution of the temperature measurement error to the observed variations. In our calculations above, we found that at a temperature error of ± 0.5 °C (median of reported uncertainties) *EFs* range from 1.1 to 1.6 for atmospheric samples, and from 1.7 to 3.2 for highly active INP species. For the highest reported temperature uncertainties of ± 1.5 °C, *EFs* can still be below a factor of 2, but may also increase to more than 10, depending on the INP activation spectrum.

When considering the differences of two INP counters, each instrument can exhibit deviations from the real INP concentration due to individual temperature errors. If these instruments have opposite biases in their temperature errors, with one measuring "too warm" and the other "too cold", error factors need to be multiplied. If four or more instruments are intercompared, as was the case in the investigated studies, it seems rather likely that at least two have such opposite biases. In such a case, differences by a factor of 2 between two intercompared instruments are generally possibly for temperature errors in the range of ± 0.5 °C.

If one instrument with a temperature bias of +1 °C is compared to another instrument with a bias of −1 °C, differences of factors up to 5 or even 10 are conceivable. Consequently, the sensitivity to activation conditions can by itself explain a good part, or in certain cases potentially all, of the observed differences in intercomparisons studies. If the assessments of temperature uncertainties are too optimistic or other unknown factors contribute to a higher temperature error, these effects likely increase. Additionally, specific data protocols in intercomparison studies which bin data in 1 °C intervals may affect the

overall level of agreement between instruments by essentially adding an artificial temperature uncertainty of up to ± 0.5 °C to parts of the data sets.

Considering these findings, one might ask whether the current level of agreement between different INP instruments measuring the same aerosol is already as good as can be expected given the existing instrumental uncertainties; or how much more consistency can be achieved if temperature uncertainties can be further reduced? Furthermore, on a more fundamental level,

how much of the observed variation of the atmospheric INP concentration is real, and how much of it is attributable to uncertainties in the activation conditions of instruments?

One hint to answering parts of these question may come from intercomparison measurements of Snomax at temperatures below about −10°C (Wex et al., 2015; DeMott et al., 2018). At this temperature range the activated fraction was found to not further increase anymore, thus temperature uncertainty should play only a minor role. Although it was observed in both studies that

the level of agreement did significantly improve by virtually eliminating temperature uncertainty as a factor, still differences of up to a factor of 3 (Wex et al., 2015) and 5 (DeMott et al., 2018) were found, pointing to other substantial unidentified factors contributing to the total uncertainty. To a similar effect, when deviations between instruments were expressed as a temperature difference (e.g., 3.5 °C to 5 °C in DeMott et al. (2024)), these can in fact be larger than what a reasonable assumption of typical temperature errors may be able to explain, even when the opposite direction of temperature bias is

considered.





Still, our calculations have shown that limiting measurement errors of the ice nucleation activation conditions is essential for reliable INP concentration measurements, especially when multiple INP counters are compared. We therefore highly recommend to diligently and conservatively characterize temperature (and ice supersaturation) uncertainties in INP instrumentation.

**Appendix A**

Table A1 compiles selected additional quantitative and qualitative statements from the reviewed intercomparison studies supplementing Table 2 and complementing the discussion in Sect. 2.

**Table A1: List of findings of the identified intercomparison studies related to the regions of highest differences, stated potential**
**reasons for the differences, and other study-specific results.**

| Study | Study-specific Findings |
|---|---|
| Hiranuma et al., 2015 | • The sample processing (e.g. dry-dispersed vs. wet suspension) has a strong effect on the INP efficiency of Illite NX<br>• Illite NX has a strong temperature dependency, especially from −18 °C to −27 °C |
| Wex et al., 2015 | • $n_m$ varies by 9 orders of magnitude, increasing sharply from −3 °C to −12 °C<br>• all Snomax proteins were activated above −12 °C |
| Burkert-Kohn et al., 2017 | • Condensation mode INPs deviated from immersion mode measurements, pointing to an instrument related effect of CFDCs (DeMott et al., 2015; Garimella et al., 2017) |
| DeMott et al., 2017 | • The largest differences appeared at both the warmer and colder temperature end<br>• Discrepancies in the INP concentration of a factor of a few or 2 to 4 °C in terms of temperature are the state of the art under favorable conditions |
| DeMott et al., 2018 | • Many tested materials showed a strong temperature dependency; temperature uncertainties were identified as a key factor determining INP deviations<br>• The discrepancies were highest in temperature ranges where the aerosol had the strongest increase in INP activity<br>• Unifying the aerosol preparation, generation and sampling in the same laboratory had a positive effect on the level of agreement<br>• Particle collection using filters for rinsing vs. direct liquid impingers had little influence |





| Study | Study-specific Findings |
|---|---|
| | • The likely ubiquitous low INP concentration regime, where uncertainties are high, is still the greatest challenge |
| Hiranuma et al., 2019 | • The observed discrepancies are significantly higher than individual instrument uncertainties |
| Knopf et al., 2021 | • Closure can be achieved under specific conditions, but when a complex aerosol composition is present, parametrizations have difficulties |
| Brasseur et al., 2022 | • DFCSs were well-correlated, with deviations at smaller temporal overlap, resulting in up to 5 °C differences in nucleation onset<br>• The observations of 1 measurement day could not be predicted by any parametrization |
| Lacher et al., 2024 | • Differences between the whole air inlet and rooftop sampling may point to a loss of supermicron INPs during inlet sampling<br>• Rooftop DFCSs showed systematically higher INP concentrations, lowering the proportion of data within a factor of 2 to 19% to 27%<br>• Agreement was observed, although instruments were used in their original configuration, i.e., with varying freezing procedures, sampling substrates, sampling and analysis protocols and cooling rates<br>• The overall agreement within a factor of 5 could indicate the suitability of modern INP techniques for process-related cloud modelling |
| DeMott et al., 2024 | • There was no clear differences between online and offline INP concentrations<br>• Not all sources of discrepancies are currently fully quantifiable, meaning that atmospheric INP concentrations seem to be uncertain by up to 1 order of magnitude |

## Appendix B

The following figures provide further details on the analysis presented in Sect. 3.



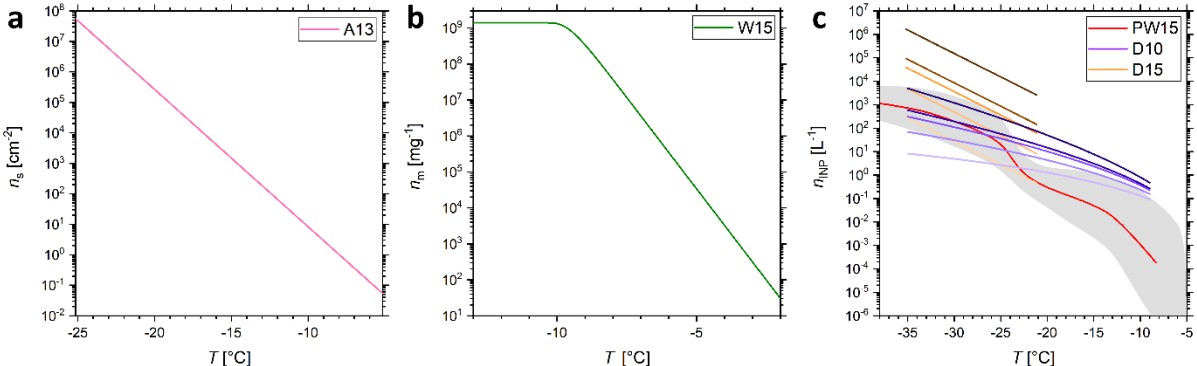

**Figure B1:** Temperature spectra of the active site density by surface $n_s$ of K-Feldspar for A13 (a). Temperature spectra of the active site density by mass $n_m$ of Snomax for W15 (b). Temperature spectra of the INP concentration $n_{INP}$ for PW15, D10 and D15 (c). The red line of PW15 is used in the analysis of the Sect. 3 as an average of the grey envelope. Note that the INP concentration per volume of water of the precipitation samples is converted to air volume as described in Petters and Wright (2015). The color scaling from light to dark for D10 and D15 represents $n_{a>0.5}$ values of 1 cm$^{-3}$, 10 cm$^{-3}$, 50 cm$^{-3}$, 100 cm$^{-3}$ and 1000 cm$^{-3}$, respectively.

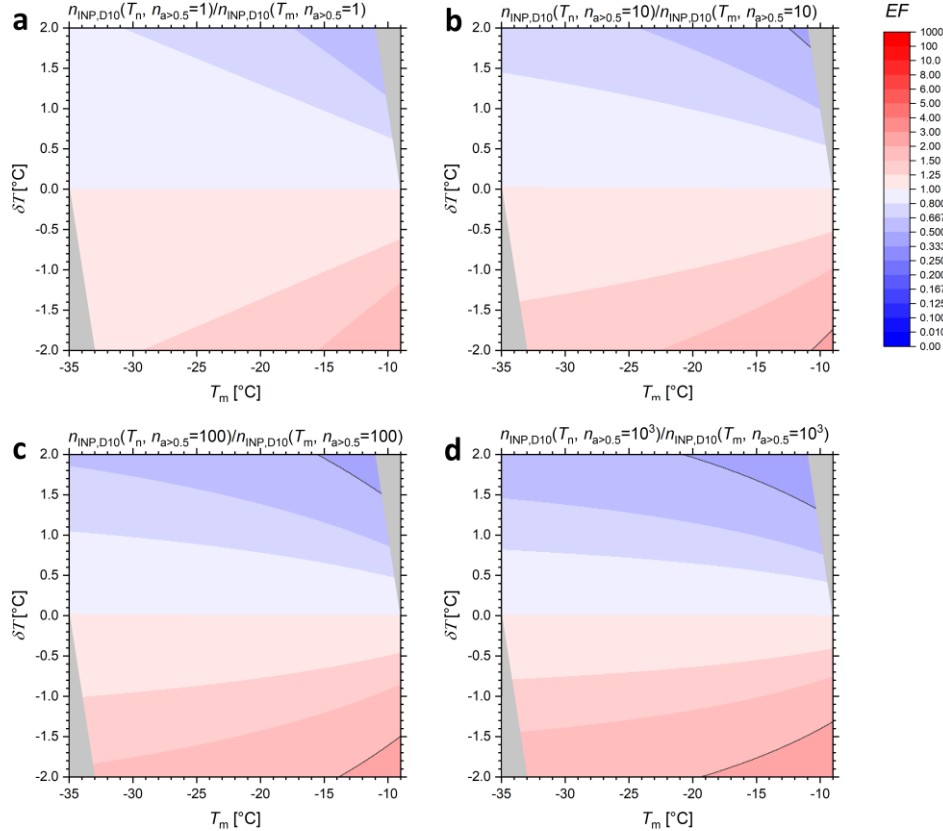

**Figure B2:** Error factors of D10 as function of assumed measurement temperature $T_m$ and temperature error $\delta T$. Labels (a) to (d) show the sensitivity to $n_{a>0.5}$ by setting $n_{a>0.5}$ to 1 cm$^{-3}$, 10 cm$^{-3}$, 100 cm$^{-3}$ and 1000 cm$^{-3}$. When visible, grey isolines show the constant error factors of 2 (and its inverse). Grey triangle overlays confine the valid range of the original data function.



**Data availability**

The data to create the figures will be uploaded to the Goethe University Data Repository (GUDe). GUDe is operated by the
University Library and the University Data Center of the Goethe University and follows the FAIR principles.

**Author contribution**

JS conceptualized the manuscript; JS reviewed the literature; JS analyzed the data and prepared the figures; JS wrote the
manuscript draft; JS and HGB revised and edited the manuscript.

**Competing interests**

The authors declare that they have no conflict of interest.

**Acknowledgements**

This research benefitted from the research infrastructure connected to the German Research Foundation projects PINEair
(DFG, SPP 1294 HALO, No 442666697) and TP-Change (DFG, TRR 301, No 428312742, project A06). We thank Joachim
Curtius for proofreading a previous version of the manuscript. We are grateful to Paul DeMott for his valuable input discussing
the content of this manuscript.

**Financial support**

This research was funded by the Deutsche Forschungsgemeinschaft (DFG, German Research Foundation) – TRR 301 –
Project-ID 428312742 (TP-Change, project A06) and by Deutsche Forschungsgemeinschaft (DFG, German Research
Foundation) – SPP 1294 HALO – Project-ID 442666697 (PINEair). The article processing charges were covered by the open-
access publication fund of the Goethe University Frankfurt, which we greatly appreciate.

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
