# Peer review of "A view on recent ice-nucleating particle intercomparison studies: Why the uncertainty of the activation temperature matters"

_EGUsphere, 2024_

## Author Response (AR1)

Response to Anonymous Referee #2

First of all, we thank the referee for submitting helpful and productive comments and annotations, which have led to improvements and clarifications within the revised manuscript, which we submit with this review response.

We have prepared a revised manuscript that addresses the questions and comments of all referees. Furthermore, below we explicitly respond to each of the items raised in the comments of anonymous referee #1. These comments are indicated in *italics,* whereas the author's response is presented in blue. Changes in the manuscript are given in green. The differences are also highlighted in separate PDFs with track changes enabled. All line and page numbers refer to the AMTD manuscript, and not the revised manuscript.
* * *
*Review of Jann Schrod and Heinz G. Bingemer, A view on recent ice-nucleating particle intercomparison studies: Why the uncertainty of the activation conditions matters*

*This manuscript, and the associated supplement, describe in detail a cause of deviation between commonly used instruments which determine the abundance of ice nucleating particles in laboratory and field studies. It has been widely noted, and specifically in recent papers in AMT and ACP, that ice nucleation measurements are critical for understanding climate, cloud formation, precipitation and a host of other atmospherically relevant issues.*

*The authors' focus is on uncertainty in temperature, which is one commonly noted cause for divergence of measurements. Although this paper is very narrowly focused, it is none the less an important addition to the literature. Several suggestions for improvement are listed below. After revisions, this paper could be suitable for publication in AMT.*

> We thank the referee for their assessment of our manuscript.

*This paper would benefit from a thorough re-write to emphasize what it is and what has been done previously.*

> We agree that the manuscript can be improved by highlighting its novel aspects. Lines 12-14 now read: "In this study, we have explored this effect as a potential reason for the differences observed among INP counters by analyzing 10 INP intercomparison studies published within the last 10 years with a novel quantitative estimate of the temperature uncertainty effect on heterogeneous ice nucleation."

> We feel however that a thorough re-write as suggested by the reviewer is not necessary, as we will argue below and in the following comments. We would like to note that the other two reviewers also did not argue for the need of an extensive rewrite in any way.

> We disagree to the reviewer's impression that we were neglectful in describing what other groups have published in regards to the importance of temperature and its uncertainty. On the contrary, we thoroughly searched the 10 identified intercomparison

papers (and other studies) for statements aiming to explain experimental differences between instruments, and in particular in regards to what role the authors ascribed to temperature uncertainty.

Here are some examples from our manuscript where we explicitly state that the effect of temperature is well-known or that explicitly list instances where temperature uncertainty has been considered in the 10 identified intercomparison studies:

- Lines 68-70: "For example, DeMott et al. (2018) points to temperature uncertainty being a key factor influencing the observed differences in INP concentration during the large-scale laboratory intercomparison of FIN-02 (Fifth International Workshop on Ice Nucleation – phase 2)."
- Lines 71-77: "The number of activated INPs is known to be very sensitive to the activation conditions, i.e. the temperature for immersion freezing, and additionally the ice supersaturation, when measurements are performed below water saturation. This fact is of course well-established and is implemented in parametrizations of specific aerosol species (e.g., Atkinson et al., 2013; Hiranuma et al., 2015; Wex et al.; 2015; Ullrich et al., 2017; Hiranuma et al., 2019) and natural atmospheric aerosol (e.g., Fletcher, 1962; DeMott et al., 2010, DeMott et al., 2015). Compilations of INP observations as presented for example in Kanji et al. (2017) or Petters and Wright (2015) also feature this finding prominently, showing a distinct exponential increase in the INP concentration with decreasing nucleation temperature."
- Lines 215-216: "As we will explore in more detail in Sect. 3, the specific slope of the temperature dependency was identified as one main cause for observed differences in some studies."
- Lines 289-292: "DeMott et al. (2024) pointed out that the INP variations between instruments measuring at Storm Peak, CO, USA significantly increased for the −20 °C to −25 °C range to nearly 2 orders of magnitude, further demonstrating the point. A similar result was also observed by Lacher et al. (2024) for the intercomparison at Puy de Dôme, France."
- Lines 384-387: "Although it was observed in both studies that the level of agreement did significantly improve by virtually eliminating temperature uncertainty as a factor, still differences of up to a factor of 3 (Wex et al., 2015) and 5 (DeMott et al., 2018) were found, pointing to other substantial unidentified factors contributing to the total uncertainty."
- Table A1

The main paragraph where we list statements made by the authors of the 10 intercomparison studies who have considered temperature effect to explain observed differences can be found prominently in the conclusions from lines 345-359. The paragraph is introduced with the words: "The strong temperature dependence of specific ice-nucleating materials combined with measurement errors of nucleation temperature were in fact considered as a partial reason for the large discrepancies observed among individual instruments in some intercomparison studies." (lines 343-345). We close the paragraph by saying: "While these examples demonstrate that the community is aware of the temperature error effect, none of these studies actually tried to quantitatively estimate the contribution of the temperature measurement error to the observed variations." (lines 360-361).

Please also see lines 138-142, 142-148, 149-150, and 157-159.

However, during the open discussion phase we were made aware of a paper by Riechers et al. (2013) who, among other things, investigated the temperature uncertainty effect for homogeneous ice nucleation. Riechers et al. (2013) also did come to very similar conclusions as we did in our analysis of heterogeneous ice nucleation. We now cite this paper in line 361:

"Riechers et al. (2013), as a notable exception, is the only previous study to our knowledge that calculated the effect of temperature uncertainty in a similarly systematic fashion, however focusing on homogeneous ice nucleation at the temperature range of 236 K – 238 K. Mirroring our results, Riechers et al. (2013) conclude that temperature accuracy is the most important factor by far for total uncertainty of the homogeneous nucleation rate, when performing an extensive error propagation including temperature accuracy, and other uncertainties related to frozen fraction increment, time interval, and radius."

*This starts with the title. The paper deals with the impact of temperature uncertainty on INP measurements. It is not a comprehensive look at all uncertainties that impact INP measurements. The title seems to imply the latter, since 'temperature' – the sole focus – isn't mentioned. I suggest "A view on recent ice-nucleating particle intercomparison studies: Why temperature uncertainty of the activation conditions matters" is actually what is done in this paper. I further suggest the authors go through and make the wording in the paper (namely abstract) consistent with their temperature uncertainty theme and focus.*

Concerning the focus on temperature and our definition of "activation conditions":

The reviewer is correct that we do focus rather narrowly on activation temperature as a potential cause for errors in INP measurements. The previous title "...Why the uncertainty of the activation conditions matters" was chosen to principally also include the uncertainty in ice supersaturation. In this sense, we defined the activation conditions as temperature for immersion freezing and additionally ice supersaturation, when measurements are performed below water saturation (lines 71-72). The ice supersaturation is in many cases probably nearly as or as important as the temperature uncertainty, which we state prominently in lines 165-168, and also for example in lines 80-87 and 159-162. Because 1) there were far less information available for $S_{ice}$ uncertainties in the literature analyzed, 2) DFCS instruments only depend on temperature due to their working principle, and 3) many INP parametrization do not account for ice supersaturation, we chose to focus in detail on temperature in the manuscript (see lines 163-165).

Concerning ambiguous phrasing of "activation conditions":

We thank the reviewer for their recommendation to look for ambiguous phrasings, and we agree that we can be clearer in some instances. Below, we list instances where we adjusted the language to better describe that only temperature was investigated and

not activation conditions in general (which could include ice supersaturation) or other parameters:

- Title: "A view on recent ice-nucleating particle intercomparison studies: Why the uncertainty of the activation temperature matters"
- Lines 28-29: "These results highlight the need to carefully assess and report on uncertainties of the ice nucleation activation temperature."
- Lines 81-82: "Consequently, comparably small changes in the activation temperature can lead to significant changes in the number of activated INPs. Although not the focus of this manuscript, similar conclusions can be drawn for the ice supersaturation."
- Lines 94-95: "The alternative would be to clearly indicate the activation uncertainties as separate temperature (or ice supersaturation) error bars corresponding to the abscissa."
- Lines 97-98: "In this study, we will focus on the magnitude, nature and a potential cause of observed discrepancies between INP instruments running in parallel by quantitatively investigating the role of inherent instrument uncertainties in the activation temperature."
- Lines 153-155: "Usually the coldest sensor at the bottom is assumed to be representative for the activation temperature and is therefore used when results are presented (e.g., Möhler et al., 2021, Knopf et al., 2021), which is a reasonable, but ultimately unverified assumption."
- Lines 314-315: "The role of uncertainties and inaccuracies of the activation temperature of experimental ice nucleation could be explored further in a quantitative manner, when results are interpreted or INP concentrations from multiple instruments are compared."
- Lines 371-372: "Consequently, the sensitivity to activation temperature can by itself explain a good part, or in certain cases potentially all, of the observed differences in intercomparisons studies."
- Lines 391-392: "Still, our calculations have shown that limiting measurement errors of the ice nucleation activation temperature is essential for reliable INP concentration measurements, especially when multiple INP counters are compared."

Concerning the Abstract:

Related to the reviewers comment on the abstract, we feel that the abstract is very clear with regards to our specific focus on temperature. Citing from the abstract: "A potential reason for these discrepancies that deserves more consideration may be related to uncertainties and errors in the temperature measurement. As the activation of INPs is a strong function of the nucleation conditions, relatively small inaccuracies in the temperature measurement may lead to significant over- or underestimations of the INP concentration. In this study, we have explored this effect as a potential reason for the differences observed among INP counters by analyzing 10 INP intercomparison studies published within the last 10 years with a novel quantitative estimate of the temperature uncertainty effect on heterogeneous ice nucleation. The stated temperature uncertainty of instruments participating in these experiments ranged from $\pm$ 0.1 °C to $\pm$ 1.5 °C,

and was most commonly specified as ± 0.5 °C. Potential deviations resulting from typical temperature errors were compared to the reported level of agreement among intercompared methods. As a measure of the potential INP error due to nucleation temperature error, we defined the temperature error factor (TEF) as the quotient of the ice nucleation activity at the actual nucleation temperature divided by the ice nucleation activity at a potentially erroneously measured temperature."

Concerning the Title:

We also agree to adjust the title. We would now have it read: "A view on recent ice-nucleating particle intercomparison studies: Why the uncertainty of the activation temperature matters" to have it align more appropriately to what is investigated in the manuscript.

Concerning other uncertainties unrelated to temperature:

Although we strongly believe that we list all kinds of potential causes of uncertainties and divergences in INP measurements throughout the manuscript (e.g., lines 48-51, 64-67, 134-182, 209-210, Tab. A1, 225-227, 319-325, 374-376, 384-390), we do not claim to present a comprehensive picture on all uncertainties impacting INP measurements. We would argue that this also cannot be inferred from the previous title of the manuscript as it specifically said "...uncertainties of activation conditions", or the current title of the manuscript for that matter. We also say in lines 221-223 that we do not think it feasible to investigate all stated uncertainties of individual instruments in the context of our manuscript. We obviously agree that there are many more general, and instrument/method specific uncertainties that can be very relevant depending on measurement specifics. See lines 322-325: "Beyond the uncertainties of the temperature measurement, there are several other, potentially large, uncertainties in ice nucleation measurements. For example, for offline INP sampling those can be related to performing the background freezing of pure water, and the handling and storage of samples in general, which naturally need to be considered as well (e.g., Polen et al., 2018; Beall et al., 2020)." Finally, near the end of the conclusions we prominently close by saying why we do not think that temperature uncertainty is able to explain away all divergences observed in the intercomparisons (lines 384-390).

*The paper has a decided tone that temperature uncertainty haven't been addressed previously. This is absolutely untrue. Essentially all, starting with even the earliest papers on INP measurement, addressed temperature uncertainty. The line in the abstract, "A regularly overlooked reason for these discrepancies may be related to uncertainties and errors in the temperature measurement." is therefore not true – temperature uncertainty isn't "overlooked", it can be found in essentially all papers on this topic. To illustrate this point, a recent intercomparison paper referenced here (DeMott et al., 2024) uses the term 'temperature' no less than 100 times and details temperature uncertainty and other factors, e.g., aerosol spreading in instrument lamina – mentioned in this paper only briefly - among a more comprehensive list of reasons for experimental divergence.*

We definitely did not intend for a reader to perceive the decided tone the reviewer describes. We regret that the reviewer feels this way. Judging from the lack of comments in this regard from the other two reviews, we think that for some sentences our

underlying intended message may have been possibly misinterpreted, while on the other hand some of our phrasing can certainly be improved by clearer language (see examples below) to avoid leaving someone with the described impression. We thank the reviewer for helping us identify those instances.

First and foremost, we like to state that we absolutely, 100%, agree that the effect of temperature uncertainty is generally well-known and that it has been considered for divergences observed in ice nucleation intercomparison studies. Having said that, there is a difference between theoretically knowing something should be important and calculating the effect that is has, as we do in the present manuscript.

As mentioned in our first comment, we already present an extensive list of examples in our manuscript where other studies specifically consider the effect of temperature and its uncertainty in their discussion of results. In fact, we have deliberately and meticulously worked through all 10 intercomparison papers to gather all instances where temperature has been considered and incorporated these into text and tables of our manuscript (see above). (As a side note, we find it obvious that a comprehensive 34 page paper investigating an intensive INP intercomparison campaign is going to have a lot of uses of the word "temperature".)

We agree that the specifically mentioned quote saying that temperature uncertainty is "regularly overlooked", was probably not phrased perfectly correct. Our intention was to say that even though researchers may know that temperature uncertainty plays a relevant role, it should be considered even more (especially when trying to explain divergences of different INP counters), as its effect can be in the order of magnitude of observed differences (as we have shown in the manuscript). We have rephrased the sentence to better represent what we intended to say: "A potential reason for these discrepancies that deserves more consideration may be related to uncertainties and errors in the temperature measurement." We give another example line (69-70), following a reference of DeMott et al. (2018) who identify temperature uncertainty as a key factor determining observed differences (lines 67-69), that mirrors our intended tone: "However, the quantitative effect of temperature uncertainty has not been fully considered in intercomparison studies thus far and deserves a more thorough investigation."

In summary, we only intended the manuscript to encourage other researchers to place more emphasis on the potential effect of temperature uncertainties and to characterize their instrumentation accordingly, potentially improving their best practices regarding temperature assessment and calibration of their instrument. We hope our final sentences in the abstract (lines 28-29), Section 3 (lines 311-312) and the conclusions (lines 391-394) do convey this intention.

Here follows a list of rephrasings, trying to improve the language as described above. We focused the rephrasings on the novel aspect of this study, namely a quantitative analysis of the role of temperature uncertainty in INP intercomparison experiments.

- Lines 10-11: "A potential reason for these discrepancies that deserves more consideration may be related to uncertainties and errors in the temperature measurement."

- Lines 69-70: "However, the quantitative effect of temperature uncertainty has not been fully considered in intercomparison studies thus far and deserves a more thorough investigation"
- Lines 88-89: "The uncertainty in the activation conditions of INP instruments are sometimes only stated briefly in the method section of respective publications, or in the supplementary material, or in rarer cases not at all."
- Lines 95-96: "However, these are often missing in figures, usually in favor of clarity, or possibly sometimes because they might have been inadvertently misjudged as not as important."
- Lines 97-98: "In this study, we will focus on the magnitude, nature and a potential cause of observed discrepancies between INP instruments running in parallel by quantitatively investigating the role of inherent instrument uncertainties in the activation temperature."
- Lines 122-124: "Although for the clear majority of cases temperature uncertainty was indicated, in more than one fifth of the cases (22/104) no estimate was given in the original study or its supplementary material."
- Lines 314-315: "The role of uncertainties and inaccuracies of the activation temperature of experimental ice nucleation should be explored further in a quantitative manner, when results are interpreted or INP concentrations from multiple instruments are compared."

Furthermore, concerning other potential reasons of experimental divergences we refer to our previous comment. Aerosol spreading in the laminar flow is mentioned in detail in our manuscript in lines 142-148 and referenced in Tab. A1.

*I suggest the authors go through and make the wording in the paper consistent with this being an extension of what has been previously discussed in INP papers on temperature uncertainty and not as if this is the only, or one of a very few, doing so. In fact, on page 4, the authors note they couldn't find sufficient temperature data in their 10 listed intercomparisons in only 21% of cases. That implies that there is significant treatment in the vast majority of cases.*

We did go through the paper as suggested by the reviewer to check the language and rephrased sentences were we thought it beneficent, as can be found in the previous replies. As mentioned earlier, we strongly disagree with the notion suggesting that we were claiming to be one of only few researchers considering temperature uncertainty.

As noted before we totally agree that in general researchers know very well that temperature uncertainty is important. Still we find that more than one fifth of researchers not stating their uncertainty in one of the main causes for total INP uncertainty in any way, shape, or form actually to be notably high and absolutely worth mentioning. We argue that it is fair to list this fact, and the frequent lack of temperature error bars in figures, as an argument demonstrating that the role of temperature uncertainty could be more strongly considered in some studies.

As listed in the above examples, we did rephrase the sentence however to: "Although for the clear majority of cases temperature uncertainty was indicated, in more than one fifth of the cases (22/104) no estimate was given in the original study or its supplementary material."

*To offer a compromise, I believe what the authors are driving at is (1) they have conducted a more thorough treatment of how temperature uncertainty can impact INP concentration - the value of this paper - and (2) that intercomparisons should provide more information, consistent with the quantities they list towards the bottom of page 3 and elsewhere in the text. This is a suggestion that can be emphasized in the abstract and conclusions (i.e., that more data should be provided). The statements that temperature uncertainty aren't addressed is, by their own references, simply not true.*

> We agree with the reviewer's assessments (1) and (2). That is exactly what the manuscript aims to deliver.

> To our understanding, there are no statements in the text claiming that temperature uncertainty is not addressed in other studies. We hope that our rephrasing of the language in some instances helps to better understand our meaning.

*In conclusion, where this paper shines is in its treatment of how much deviation can be expected from a given level of temperature uncertainty. This is an important addition to the literature and is worthy of publication. The paper suffers from is the incorrect and overly strong statements that temperature is not treated in previous papers. If the latter can be eliminated this work will be much stronger.*

> We thank the reviewer for their praise.

> Once again, we disagree with the reviewer's statements implying that we were claiming that temperature is not treated in previous studies. As indicated earlier, we softened and rephrased the language where we thought necessary to avoid leaving the described impression.

> We thank the reviewer for helping us identify some of these possibly too strongly phrased statements and also think that the paper has improved due to these clarifications.

Response to Anonymous Referee #3

First of all, we thank the referee for submitting helpful and productive comments and annotations, which have led to improvements and clarifications within the revised manuscript, which we submit with this review response.

We have prepared a revised manuscript that addresses the questions and comments of all referees. Furthermore, below we explicitly respond to each of the items raised in the comments of anonymous referee #1. These comments are indicated in *italics,* whereas the author's response is presented in blue. Changes in the manuscript are given in green. The differences are also highlighted in separate PDFs with track changes enabled. All line and page numbers refer to the AMTD manuscript, and not the revised manuscript.
* * *
*Review of Schrod et al. A view on recent ice-nucleating particle intercomparison studies: Why the uncertainty of the activation conditions matters*

*This manuscript addresses the uncertainties associated with in situ ice nucleation measurements based on a review of a number of intercomparison studies made over the last ten years. It illustrates how temperature uncertainties can be the result of the large spread in the range of ice nucleation measurements. They calculate these estimated uncertainties as a function of temperature and determine an error Factor (EF). This EF is then evaluated using a number of commonly used parametrizations from the literature. This paper is well written and pleasant to read. It is a very important topic to address and this work will likely contribute to the motivation to generating new measurement guidelines and approaches when comparing instruments.*

> We thank the reviewer for their assessment of our manuscript. We do hope that researchers will consider our findings in upcoming instrument intercomparisons or instrument characterizations in general.

*However at the end of the manuscript, the impression is that, the current uncertainty in ice nucleation measurements is so great that we cannot rely on these measurements when interpreting particle ice-nucleating properties.*

> This is not what we aimed to relay as the message of the manuscript. Rather, we mainly wanted to make researcher more aware of potential uncertainties in their measurement data (e.g., see last paragraph). If temperature uncertainties are large, measurement errors may impede the reliability of the INP data. We hope that our raised questions did not come off as too negative. In fact, we feel that the community is moving forward in great strides, considering the development of new instruments, a larger coverage of observational data in space and time, and more consistent intercomparisons, which we also note in lines 210-211. We still think that raising questions can help the community identify problems and move towards answering unresolved issues.

*It would be a useful addition to this manuscript to include a list of recommendations that can be brought forward into future measurements. The community is already striving to reduce the uncertainties in the measurements.*

- *Are there some methods that have shown to have smaller uncertainties and more reliable measurements?*
- *Should the community compare similar instruments (same make/model) and avoid comparing different types of ice nucleating measurements?*
- *How can these temperature measurements be improved?*

Adding a list of recommendations could be a great idea. Our final sentence does already recommend to diligently and conservatively characterize temperature (and ice supersaturation) uncertainties in INP instrumentation, which, as the reviewer correctly says, the community of course is already trying to accomplish anyway. Although interesting questions for sure, we don't feel that the three suggested bullet points are a good fit to finishing off of the manuscript however, and we prefer to reply here to reviewer 2 instead.

- We do not intend to evaluate the performance of individual methods or instruments. What we can say from our literature analysis is what is presented in Fig. 1 and Tab. S1: Generally, researchers stated lower temperature uncertainties for DFCS instruments. As we have listed in Section 2, there are arguments that the temperature (and supersaturation) uncertainty may be larger in online INP instruments. As all instruments have strengths and weaknesses, we do not think, however, that one type of instrument is more reliable than another. Having a plethora of methods seems like a great way to tackle the complex study of atmospheric ice nucleation.
- We think that there is merit for both suggestions. When comparing two very similar instruments (maybe even of the same make), you could focus on small details of the general performance, activation conditions measurements, INP counting algorithms, etc. Comparing multiple different methods is also very valuable, when it is assured that they measure the same thing (i.e., both immersion freezing INP concentration at -25°C and water saturation). Consistency of results among multiple independent methods with different working principles gives good confidence in the reliability of the data. Also the strengths (e.g., explorable temperature regime, time resolution) of one instrument may complement the others weaknesses and vice versa.
- We feel that this question is difficult to answer in a general way, as it very much depends on individual instrument specifics. Thus individual research group would know best, how to improve their own instrument. The best advice we can give generally, is what researchers will already know: Make as much measurements of the activation conditions as possible. Make sure that your measurement best represent the conditions where the ice crystals are formed. Do frequent calibrations of your sensors. Be conservative, when estimating uncertainties.

Response to Anonymous Referee #1

First of all, we thank the referee for submitting helpful and productive comments and annotations, which have led to improvements and clarifications within the revised manuscript, which we submit with this review response.

We have prepared a revised manuscript that addresses the questions and comments of all referees. Furthermore, below we explicitly respond to each of the items raised in the comments of anonymous referee #1. These comments are indicated in *italics,* whereas the author's response is presented in blue. Changes in the manuscript are given in green. The differences are also highlighted in separate PDFs with track changes enabled. All line and page numbers refer to the AMTD manuscript, and not the revised manuscript.
* * *
*The submitted manuscript, "A view on recent ice-nucleating particle intercomparison studies: Why the uncertainty of the activation conditions matters," from Schrod and Bingemer, is a detailed investigation into the potential consequences of temperature uncertainties associated with various measurement techniques, targeted at determining ice nucleating particle (INP) concentrations. I find the manuscript well written, fairly easy to digest, and also note that it raises an important point of discussion for the community participating in these measurements. Although the title is somewhat general, the submitted manuscript really only deals in detail with temperature uncertainty. As the authors point out, even though many investigators discuss and consider temperature uncertainty, the common practice is that results like activation curves are not reported with $\Delta T$ error bars. I believe that the author's main point is that, especially when activation curves are steep, small temperature changes mean pronounced changes in INP activity parameters. Although there are several technical corrections needed throughout the manuscript, I would suggest that with those corrections this is a suitable manuscript for publication and a valuable contribution to the field. I suggest points for technical correction in (page number, line number) form below.*

We thank the reviewer for their assessment of our manuscript.

In accordance to the comments of reviewer 1 we rephrased the title to "A view on recent ice-nucleating particle intercomparison studies: Why the uncertainty of the activation temperature matters" to more fittingly represent the focus of the manuscript.

As reviewer 1 suggested, we will more prominently state that most investigators do consider temperature uncertainty, although we feel that we have done so already on several instances throughout the text.

We will go through the suggestions of technical corrections one by one.

*Itemized technical corrections:*

*Abstract -- The authors should consider (for the atmospheric community) if they would like to make a more clear distinction between their choice of EF to mean "error function", given this is also a common notation for "emission factor".*

*Yes, the reviewer is absolutely correct. To avoid any confusion, we now say temperature error factor (TEF) throughout the text.*

*(page 2, line 36) -- strike "out", should simply read, " ....ice particles precipitate earlier..."*

Corrected as suggested.

*(3,77) "Per" is strange to begin a sentence with, I would suggest, "For every 5..."*

Corrected as suggested.

*(3,80) replace "highest" with "most" and "aerosol" with "particles"*

Corrected as suggested.

*(3,85) strike "according to Eq. (1) and Eq. (2):" and replace with "as"*

Corrected as suggested.

*(3,88) should be "methods sections"*

Corrected as suggested.

*(4,95) I suggest replacing "of" with "corresponding to the"*

Corrected as suggested.

*(4,103) Rephrase: We considered studies for our investigation when the following criteria are all met:*

Corrected as suggested.

*(4,104) specify "published since 2015" as in the last 10 years will not age well.*

Corrected as suggested.

*(4, 109) rephrase, "We identified..."*

Corrected as suggested.

*(4, 111) strike "a few"*

Corrected as suggested.

*(4,124) The sentence beginning, "A number of uncertainty estimates...." needs to be clarified and/or expanded. What kind of interpretation?*

What we mean is that some statements were not in the style of "The temperature uncertainty is ± X.X °C". Instead, we have a long list of footnotes (a to l) in Table. S1 in the supplement (to which we know refer in the listed sentence), highlighting the cases

where estimates were somewhat ambiguous. Of those, the most commonly observed non-straight-forward estimate was when the text was a little unclear whether accuracy or precision is indicated. For another example, sometimes the authors give multiple error sources and the reader needs to decide if those can simply be added together. Or sometimes only a minimal temperature error or an uncertainty range is stated. Also there was one instance where one measurement was described at a temperature of -15 °C ± 1.5 °C. The text does not explain however if this reflects temperature variation during a specific experiment or uncertainty.

As we further explain in lines 128 and following, even if clear statements were made, often no description was given detailing how this estimate came about. Further, when the estimate was not given in the study itself, it was sometimes not easy to decide which of several other estimates to choose from of those other studies using the same specific instrument. Finally, there sometimes exist different versions of the same instrument at different institutes and they give different temperature uncertainty estimates.

We did change "A number of" to "Some" however.

*(5, 140) -- When discussing the Castarède et al., 2023, paper I would suggest the authors also highlight that in this manuscript the authors make some effort to argue that in CFDCs (in particular, PINCii there) it might be that the important activation condition is in fact the strongest thermodynamic forcing condition present within the chamber at a given time. The details appear to be discussed more in depth in the first author's PhD thesis. But this raises an important issue with online type instruments. At times activation conditions are the important reported parameter, not simply INP counting. The two types of measurements will not be impacted in the same way by temperature uncertainty of such chambers.*

We added a reference of Castarède and Brasseur et al. (2023) to the sentence in lines 137-138. Further, we agree with the reasoning that activation conditions are often the most important parameter influencing the INP concentration estimate and often outweigh simply INP counting uncertainties. We have said so in lines 225-227 and hint at in lines 331-334 for example.

*(6, 175) The range of temperatures is strangely presented. Mixing digits and text, and "tenth" should at a minimum be "tenths" I believe.*

Corrected as suggested.

*(8, 198) suggest: ....different instruments usually agree to within 1 order....*

Corrected as suggested.

*Table 2: $n_m$ is introduced in table without first being defined in text. Also in the first bullet related to the DeMott et al., 2017 paper, it is not clear weather differences get smaller or larger as concentrations go above or below 1 INP/L. Please rephrase so intent is clear.*

Actually, $n_m$ is first introduced in page 3 line 91, thus prior to Table 2. Differences in DeMott et al. (2017) increase at lower INP concentrations. We rephrased the text to make it clear.

*(10, 247) perhaps: ...density per unit surface...*

> Corrected as suggested.

*(11, 260) "upper grey line" and in fact all of the "grey isolines" referred to in the caption are extremely difficult to distinguish.*

> We prepared a new version of Fig. 2, with more easily distinguishable isolines, changing their color to a bright yellow with more contrast.

*(11, 262) This claim that "a hypothetical instrument mistakenly assuming to measure at -25 would actually report nINP of the true temperature of -23.5 is difficult to visualize with the presented figures. Can the reader be coached through how to understand this?*

> We understand that it is difficult to intuitively understand Fig. 2. Therefore, we did add the hypothetical scenario, which, it seems, still does not fully deliver in helping to understand what is depicted. We try again in other words here: Figure 2 plots the actual temperature error (likely unknown to the researcher) versus the measured instrument temperature. The resulting over- or underestimation due to a false temperature measurement is presented in the color code (and isolines). The hypothetical scenario means to illustrate this. Here, at the reported temperature of -25°C a concentration of 25 $L^{-1}$ is measured (cannot be inferred from Fig. 2, but can be calculated from Eq. 4). However, in this scenario the actual temperature of the instrument was -23.5°C (i.e., +1.5°C warmer). At the intersect of the incorrectly measured temperature (-25°C) and a +1.5°C temperature error, Fig. 2d gives a TEF of 0.5. This means that the instrument reports a value that is 50% lower than what the real concentration at -25°C would have been (50 $L^{-1}$ in this example). Or in other words, when you assume that your temperature measurement is absolutely correct, but the actual activation temperature was 1.5°C warmer than you assumed, your reported INP concentration is 50% too low. We have added a circle to Fig. 2d to mark the intersect of $T_{m}$ = -25°C and $\delta T$ = +1.5°C. We hope that with this clarification and the adjustment to Fig. 2 it is now understandable.

*(12, 282) suggest: $\delta T$ is larger*

> Corrected as suggested.

*(12, 293) suggest: ...is acceptable, increasing only to an EF...*

> Corrected as suggested.

*Conclusions: I think for the offline droplet/assay freezing methods the fact that time dependence is largely ignored needs to be mentioned again (as I believe it is in the introduction) in the paragraph spanning pages 13 and 14.*

> We added a sentence to line 325: "Furthermore, it should be mentioned that the time dependency of nucleation events is often disregarded in DFCS measurements."

*(15, 369) suggest "possibly" should be replaced by "possible"*

Corrected as suggested.

*(15, 380) This is a great question, and I applaud the authors trying to take one step to solving this underlying problem.*

We thank the reviewer for their praise. We hope that our raised questions did not come off as too negative. In fact, we feel that the community is moving forward in great strides, considering the development of new instruments, a larger coverage of observational data in space and time, and more consistent intercomparisons, which we also note in lines 210-211.

*Appendix B: suggest: The following figures provide further details for (or perhaps from) the analysis presented in....*

Corrected as suggested.